# Accuracy of heart rate measurement using AirPods Pro 3 during graded treadmill exercise: A laboratory-based validation study

Cailbhe Doherty[1,2]*, David Burke[3,4], Owen Mitchell[1], David Lawless[1], Jasman Brar[1], Michael Fuchs[1], Rory Lambe[1,2]

1 School of Public Health, Physiotherapy and Sports Science, University College Dublin, Dublin, Ireland, 2 Insight Research Ireland Centre for Data Analytics, University College Dublin, Dublin, Ireland, 3 Cardiology, Beacon Hospital, Dublin, Ireland, 4 School of Medicine, University College Dublin, Dublin, Ireland

* cailbhe.doherty@ucd.ie

## Abstract

AirPods Pro 3 incorporate an in-ear optical heart-rate sensor, but independent validation during exercise is limited. We evaluated agreement between AirPods Pro 3 heart rate and an ECG-derived chest-strap reference (Polar H10) during graded treadmill exercise in a controlled laboratory setting. Forty adults (mean age 23·8 years; 37·5% female) completed a protocol comprising rest and progressive exercise stages targeting ~40–85% of age-predicted maximal heart rate, including rapid workload transitions. Heart-rate time series from both devices were synchronised by timestamp and aggregated into non-overlapping 5-s epochs. Agreement was assessed using a repeated-measures Bland–Altman approach implemented via a linear mixed-effects model with participant-level random effects; absolute error metrics were calculated at the participant level and summarised overall and by intensity category. Across 16,735 paired epochs, mean bias was −0·03 beats·min⁻¹ (AirPods Pro 3 minus Polar H10; 95% CI −0·22 to 0·17), indicating negligible systematic error. The total standard deviation of differences was 5·23 beats·min⁻¹, yielding 95% limits of agreement from −10·27 to 10·22 beats·min⁻¹, with greater dispersion at higher heart rates. Overall mean absolute error was 2·08 beats·min⁻¹ and mean absolute percentage error was 2·02%, with mean absolute error ranging from 1·31 to 2·4 beats·min⁻¹ across intensity categories. AirPods Pro 3 therefore provided heart-rate estimates closely aligned with a validated chest-worn reference during graded treadmill exercise in healthy adults, with minimal bias and low average error but wider epoch-to-epoch variability at higher intensities.

## Author summary

Many people use consumer devices to monitor heart rate during exercise and daily life, but most validation studies have focused on wrist-worn watches.

---

**Data availability statement:** The full analysis code is publicly available at https://github.com/rorylambe/airpods. The minimal anonymized dataset necessary to replicate the study findings is publicly available at https://doi.org/10.17605/OSF.IO/D4JNS.

**Funding:** This work was supported by the Health Research Board Ireland (HRB ILP-PHR-2024-005 to CD) and Research Ireland (12/RC/2289 P2 to CD). The funders had no role in study design, data collection and analysis, decision to publish, or preparation of the manuscript.

**Competing interests:** The authors have declared that no competing interests exist.

Ear-worn devices are newer, and independent evidence on their accuracy during exercise is limited. In this study, we evaluated whether AirPods Pro 3 can measure heart rate accurately during treadmill exercise.

Healthy adults completed a structured laboratory protocol including rest, walking, running, and recovery. Participants wore AirPods Pro 3 together with a validated chest-worn heart rate monitor, which served as the reference standard. We compared the heart rate values recorded by the two devices across thousands of short time intervals spanning different exercise intensities.

AirPods Pro 3 showed very small average differences from the chest-worn monitor, with no meaningful overall tendency to overestimate or underestimate heart rate. Average error remained low across rest, light, moderate, and vigorous exercise. Variability increased at higher intensities, meaning that individual readings could differ more during harder exercise.

These findings suggest that AirPods Pro 3 can provide useful heart rate information for fitness tracking in healthy adults, but they should not replace clinical-grade equipment when high precision is required.

## Introduction

Consumer wearables are body-worn digital devices designed to capture physiological and behavioural data during everyday life. Over the past decade, these devices have expanded in scope and adoption, enabling large-scale measurement of physical activity, sleep, and cardiovascular signals [1]. Common form factors include finger-worn devices such as smart rings (e.g., those produced by Oura Health), strap-based fitness bands (e.g., WHOOP), and wrist-worn smartwatches (e.g., Apple Watch, Fitbit) [1]. More recently, ear-worn devices have emerged as a distinct category of consumer wearable capable of supporting biometric sensing [2]. Unlike wrist- or finger-based systems, ear-worn devices are positioned closer to central circulation and are less exposed to peripheral vasoconstriction, introducing a distinct anatomical context for physiological measurement within the broader wearable ecosystem [2,3].

Apple AirPods are among the most widely adopted true wireless stereo earbuds globally, functioning both as audio accessories and as body-worn consumer devices [4]. By 2024, Apple sold an estimated 66 million units of AirPods worldwide [5], and the product line continues to anchor a substantial share of the wireless earbud market, with analysts estimating that AirPods accounted for roughly 23–31% of global wireless earbud shipments in recent years [6]. This level of adoption confers public health relevance on any biometric sensing capability embedded within the platform, because even modest systematic measurement error could affect large numbers of users and shape digital health monitoring behaviours at scale.

The latest generation, AirPods Pro 3, incorporates an in-ear optical heart-rate sensor that supports continuous biometric measurement during physical

activity. The sensor uses pulsed infrared light to detect changes in optical absorption associated with blood flow within the ear canal, following principles analogous to photoplethysmography (PPG) [7]. To improve robustness during movement, optical signals are integrated with inertial data from onboard accelerometers and gyroscopes, alongside contextual information from the paired smartphone and an on-device machine-learning model [7]. This multimodal approach aims to provide real-time heart-rate estimates during exercise without requiring a separate chest strap or wrist-worn device. Despite their integration within a broader consumer health ecosystem, the performance of this ear-based optical implementation has not been independently evaluated under controlled exercise conditions.

Accurate heart-rate measurement is central to both individual- and population-level health assessment because it reflects underlying autonomic and cardiovascular function [8,9]. At rest, heart rate provides an accessible marker of cardiometabolic health, with deviations from expected ranges associated with increased risk of hypertension, coronary disease, and mortality [10,11]. During physical activity, heart rate serves as a practical indicator of exercise intensity and physiological strain, informing training prescription, recovery assessment, and daily load management [12,13]. When combined with contextual information such as activity duration and intensity, heart-rate data also support estimation of derived metrics, including maximal aerobic capacity (VO$_2$ max), which is a strong predictor of cardiovascular and all-cause mortality [14,15].

These applications underscore the importance of understanding the accuracy of wearable-derived heart-rate measurements in settings relevant to real-world use. Heart rate is a dynamic biological signal that fluctuates rapidly in response to changes in posture, movement, workload, and autonomic regulation, making it particularly susceptible to measurement error during physical activity [16]. Optical sensing approaches are vulnerable to multiple sources of artefact, including motion-induced sensor displacement, deformation of skin and underlying tissue, muscle contractions, and changes in local blood flow [16,17]. These challenges are amplified during conditions that are most relevant to real-world use, such as high-intensity exercise, rapid or intermittent transitions in workload, and activities involving substantial upper-body movement. Validation studies that rely primarily on averaged values or steady-state protocols may therefore underestimate error and fail to characterise performance under conditions that reflect intended use [16,17].

In response to these challenges, contemporary frameworks for validating wearable heart-rate sensors emphasise evaluation under both steady-state and dynamically changing conditions. The INTERLIVE consortium recommends protocols that span a wide range of exercise intensities, include rapid transitions in physiological demand, and preserve the temporal structure of the heart-rate signal rather than relying solely on summary averages [16]. When repeated measurements are obtained from the same individual, agreement is most appropriately assessed using repeated-measures approaches, such as Bland–Altman analyses implemented within mixed-effects models, which account for within-participant dependence and time-varying variability [16]. Epoch-based analyses further balance temporal resolution with statistical stability and are particularly suited to dynamic exercise protocols. Together, these principles define current best practice for evaluating wearable heart-rate accuracy under conditions that reflect intended use.

The aim of this study was to evaluate the agreement between heart-rate measurements obtained from AirPods Pro 3 and a validated chest-worn reference during graded treadmill exercise, with particular emphasis on performance across varying exercise intensities and during dynamic transitions in heart rate.

## Results

### Participant characteristics and analytical sample

Forty participants completed the laboratory protocol and were included in the primary analysis. Baseline characteristics are summarised in Table 1.

**Table 1. Baseline characteristics of participants (N = 40).**

| Characteristic | Value |
|---|---|
| Age, mean (SD), years | 23.8 (3.6) |
| Age range, years | 18–37 |
| Female, n (%) | 15 (37.5) |
| Height, mean (SD), cm | 175.4 (10.4) |
| Weight, mean (SD), kg | 76.2 (14.3) |
| Body mass index, mean (SD), kg/m² | 24.6 (3.0) |
| Body mass index range, kg/m² | 20.2–32.2 |
| Fitzpatrick skin type | |
| Type I | 6 |
| Type II | 2 |
| Type III | 14 |
| Type IV | 18 |

## Primary agreement analysis

Overall agreement between AirPods Pro 3 and the Polar H10 is illustrated in the repeated-measures Bland–Altman plot (Fig 1). Across 16,735 paired epochs, AirPods Pro 3 showed minimal systematic bias relative to the Polar H10 device. The mean difference was −0.03 beats·min⁻¹ (AirPods Pro 3 minus Polar H10; 95% CI −0.22 to 0.17).

Between-participant variance was small ($\tau^2 = 0.61$), whereas within-participant variance was larger ($\sigma^2 = 26.73$). This pattern indicates that most variability arose from epoch-to-epoch fluctuations rather than systematic differences between individuals. The resulting total standard deviation of differences was 5.23 beats·min⁻¹, yielding 95% limits of agreement from −10.27 to 10.22 beats·min⁻¹. Bootstrap-derived confidence intervals ranged from −14.48 to −6.55 beats·min⁻¹ for the lower limit and from 6.32 to 14.65 beats·min⁻¹ for the upper limit.

Visual inspection of the Bland–Altman plot showed no evidence of systematic proportional bias. However, dispersion of differences increased at higher mean heart rates.

## Agreement by exercise intensity

Complementary absolute error metrics stratified by exercise intensity are summarised in Table 2. Across all intensity categories, MAE was low, ranging from 1·31 beats·min⁻¹ at rest to 2·4 beats·min⁻¹ during vigorous-intensity exercise. MAPE ranged from 1·52% during vigorous exercise to 2·32% during light-intensity activity. Participant-averaged signed differences varied modestly across intensity zones, with small positive bias at rest and moderate intensities and a small negative bias during light and vigorous exercise. Although variability in epoch-level differences increased with exercise intensity (as reflected by larger standard deviations at higher heart rates) average absolute error remained stable across light, moderate, and vigorous workloads (approximately 2·2–2·4 beats·min⁻¹).

## Discussion

The aim of this laboratory-based validation study was to evaluate the agreement between heart-rate measurements obtained from AirPods Pro 3 and a validated chest-worn reference. Our findings showed that under dynamically changing physiological conditions, AirPods Pro 3 showed close agreement with the Polar H10, a chest-worn, ECG-derived reference during graded treadmill exercise. Across forty participants and 16735 paired epochs, systematic bias was negligible, with a mean difference of −0·03 beats·min⁻¹ (95% CI −0·22 to 0·17). Average error was low, with a mean absolute error of 2·08 beats·min⁻¹ and a mean absolute percentage error of 2·02%. These findings indicate that AirPods Pro 3 can track

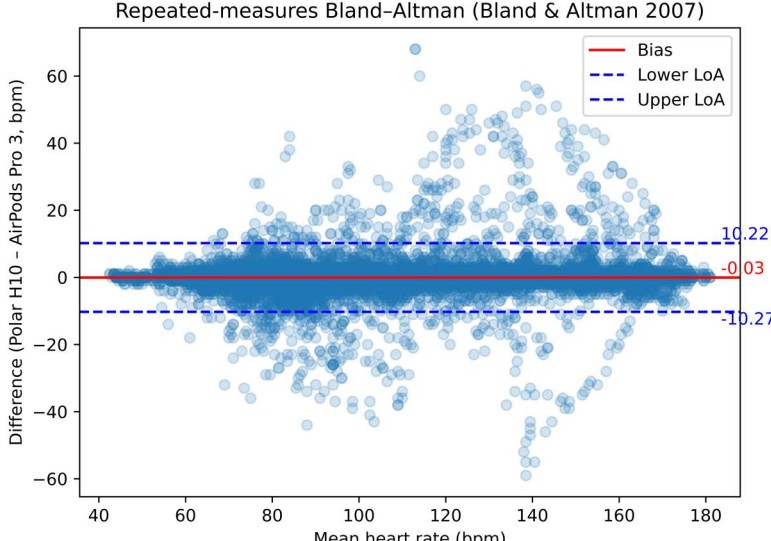

**Fig 1. Repeated-measures Bland–Altman plot showing agreement between AirPods Pro 3 and Polar H10 across five-second epochs.**

**Table 2. Agreement between AirPods Pro 3 and Polar H10 for heart rate, stratified by exercise intensity.**

| Intensity zone | %HR$_{max}$ range | Paired epochs, n | Mean heart rate (AirPods Pro), beats·min$^{-1}$ | Mean heart rate (Polar H10), beats·min$^{-1}$ | Mean bias*, beats·min$^{-1}$ | SD of differences, beats·min$^{-1}$ | Mean absolute error, beats·min$^{-1}$ | Mean absolute percentage error, % |
|---|---|---|---|---|---|---|---|---|
| Rest | <40% HR$_{max}$ | 3435 | 67·7 | 67·6 | 0·09 | 2·79 | 1·31 | 1·9 |
| Light | 40–59% HR$_{max}$ | 7830 | 96·5 | 96·6 | -0·11 | 5·56 | 2·24 | 2·32 |
| Moderate | 60–74% HR$_{max}$ | 3414 | 131·7 | 131·4 | 0·35 | 5·78 | 2·3 | 1·76 |
| Vigorous | ≥75% HR$_{max}$ | 2056 | 159·1 | 159·6 | −0·53 | 5·97 | 2·4 | 1·52 |

*Mean bias calculated as AirPods Pro 3 minus Polar H10.

heart-rate dynamics with good accuracy across a wide range of workloads, including during rapid transitions in physiological demand. However, while mean bias and average error were small, variability in heart-rate differences increased with exercise intensity; the total standard deviation of differences was 5·23 beats·min$^{-1}$, with 95% limits of agreement ranging from −10·27 to 10·22 beats·min$^{-1}$. Importantly, mean absolute error remained consistently low across rest, light, moderate, and vigorous intensity categories (approximately 1·3–2·4 beats·min$^{-1}$). This combination of minimal systematic bias with widening dispersion at higher workloads is consistent with the known physiological and biomechanical challenges of optical sensing during movement-intensive and non–steady-state exercise [16].

Interpretation of these agreement metrics depends on the intended use of heart-rate data. Limits of agreement approaching ±10 beats·min$^{-1}$ are comparable to those reported for other consumer optical wearables evaluated under dynamic exercise conditions [1,18] and are unlikely to materially affect applications such as population-level surveillance, behavioural feedback, or longitudinal self-monitoring, where relative change and trend fidelity are often more informative than momentary precision [1,8]. At the individual level, the low mean absolute and percentage errors indicate that AirPods Pro 3 captured central tendencies in heart-rate response with reasonable accuracy across exercise intensities.

 

However, the observed dispersion constrains applications that require fine discrimination between adjacent training zones or detection of brief, transient physiological excursions. On this basis, these findings support the use of ear-based optical heart-rate monitoring for tracking patterns and trajectories of cardiovascular effort over time, while underscoring that such devices should not be considered interchangeable with ECG-derived systems for tasks requiring beat-level precision or clinical decision-making. Systematic bias remained minimal across rest, light, moderate, and vigorous intensity categories, with no evidence of directional drift at higher heart rates. Error metrics were broadly similar across intensity zones, indicating that central accuracy was preserved as workload increased, although dispersion widened at higher workloads. Concurrently, dispersion widened at higher intensities, a pattern that is physiologically plausible given greater head, jaw, and upper-body motion, increased cardiac output and pulse pressure, and more rapid autonomic fluctuations during workload transitions. These conditions are known to amplify motion artefact and challenge optical sensing [8,16,19]. Importantly, the absence of directional drift across intensities suggests that the device remained responsive to dynamic changes in heart rate, preserving the temporal structure of the signal during both graded progression and rapid transitions rather than performing well only under steady-state conditions.

The magnitude and structure of agreement we observed are consistent with a substantial body of validation literature evaluating consumer wearable heart-rate monitors under controlled exercise conditions. Across wrist-worn optical devices, repeated independent evaluations have shown that mean bias is typically small—often within ±1–2 beats·min$^{-1}$—while variability increases with movement intensity and physiological demand. A recent living systematic review and meta-analysis encompassing 82 studies and more than 430,000 participants reported a pooled mean bias of −0·27 beats·min$^{-1}$ for Apple Watch heart-rate measurements relative to ECG or chest-strap references, with 95% limits of agreement on the order of ±7 beats·min$^{-1}$ [18]. These findings indicate high average accuracy but underscore that individual measurements can deviate meaningfully from reference values, particularly during exercise. Meta-analyses and systematic reviews spanning multiple manufacturers similarly conclude that wrist-worn PPG devices achieve acceptable validity for heart-rate measurement in laboratory settings, with MAE commonly below 5 beats·min$^{-1}$ or 5% under steady or moderately dynamic conditions [20]. However, agreement varies substantially by device model, activity type, and intensity, and performance consistently degrades as exercise intensity increases or movement becomes more irregular. This pattern has been observed across walking, running, and interval-based protocols, reinforcing that optical heart-rate sensing is inherently more susceptible to motion artefact and signal instability during vigorous or non-steady-state exercise.

Several multi-device studies further illustrate this context-dependence. Shcherbina and colleagues evaluated seven wrist-worn wearables against 12-lead ECG across rest, treadmill exercise, and cycling, reporting median heart-rate errors below 5% for six devices, but with greater error during walking and running than during cycling, where arm movement is constrained [21]. Similarly, Gillinov et al. demonstrated that wrist-worn optical monitors performed well during treadmill exercise (concordance coefficients often exceeding 0·9) but showed reduced agreement during exercise modalities involving pronounced arm motion, such as elliptical training [22]. These studies collectively highlight that widening limits of agreement at higher workloads are a reproducible feature of optical heart-rate monitoring rather than an idiosyncratic failure of individual devices.

Against this backdrop, the limits of agreement observed for AirPods Pro 3 (approaching ±10 beats·min$^{-1}$) fall within the range reported for contemporary consumer wearables evaluated during dynamic exercise. While somewhat wider than pooled estimates reported for wrist-worn Apple Watch devices under mixed conditions [18], they are comparable to those documented in studies explicitly challenging devices with graded treadmill protocols, rapid workload transitions, or high-intensity exercise [23]. Importantly, mean absolute error in the present study (~2 beats·min$^{-1}$) aligns closely with values reported for high-performing wrist-worn devices in both healthy adults and selected clinical populations [24], indicating that average deviation from the reference remained low despite increased dispersion at higher heart rates.

Chest-worn monitors, by contrast, consistently demonstrate near-perfect agreement with ECG, with concordance coefficients approaching 0·99 and mean absolute percentage errors typically below 2–3% across a wide range of activities

[22,25,26]. Their superior performance reflects fundamental differences in sensing modality: electrical detection of cardiac depolarisation is largely immune to motion artefact that challenges optical approaches. For this reason, chest straps remain the preferred reference standard in validation studies and are recommended when precise heart-rate measurement is needed for clinical decision-making or high-resolution physiological analysis [25]. Evidence specific to ear-worn heart-rate sensing remains comparatively sparse, making direct contextualisation more difficult. Early studies of prototype in-ear PPG sensors demonstrated low mean error but wide limits of agreement, leading authors to characterise the technology as "accurate but imprecise" under certain conditions [27]. More recently, validation of commercial earbud-based devices has suggested that ear-worn sensors can achieve accuracy comparable to high-end wrist wearables, particularly during cycling or other activities with limited upper-limb motion [28]. In one study of consumer earbuds evaluated against a Polar H10 reference, mean bias was negligible and mean absolute error approximated 2 beats·min$^{-1}$, although variability increased during high-intensity intervals [28].

This pattern—negligible systematic bias, low average error, and increasing variability at higher intensities—is physiologically and technically plausible given the anatomical characteristics of the ear canal and the known behaviour of optical heart-rate sensing during exercise. Compared with the wrist, the ear represents a measurement site closer to central circulation, with relatively stable perfusion supplied by branches of the external carotid artery and reduced exposure to peripheral vasoconstriction during exercise [29,30]. Optical signals obtained from this region therefore tend to exhibit higher pulse amplitude and less sensitivity to temperature- or perfusion-related fluctuations than signals acquired at distal sites such as the wrist or finger [31,32]. These features provide a plausible explanation for the negligible overall bias and low mean absolute error observed across exercise intensities in the present study. Furthermore, the widening dispersion of heart-rate differences at higher workloads is consistent with well-described limitations of PPG under conditions of substantial movement and rapidly changing physiology. Although the ear canal is comparatively protected from large limb movements, it is not motion-invariant. Running and high-intensity exercise introduce head acceleration, jaw motion, and subtle changes in earbud position that can transiently disrupt optical coupling between the sensor and tissue, amplifying motion artefact [33,34]. Increased cardiac output and pulse pressure at higher heart rates may further accentuate beat-to-beat variability in the optical signal, particularly during rapid autonomic transitions when heart rate accelerates or decelerates over short time scales [34]. Combined, these factors provide a potential explanation for the observed heteroscedasticity without invoking systematic device failure.

The technical architecture of contemporary consumer wearables also likely contributes to the observed pattern of low average error but increased variability at extremes. Optical heart-rate estimates provided to end users typically reflect the output of proprietary algorithms that integrate raw optical signals with inertial data and apply filtering or temporal smoothing to suppress noise [23,35,36]. Such approaches can stabilise mean estimates and reduce spurious outliers but may also attenuate responsiveness to abrupt changes in heart rate, leading to transient discrepancies relative to ECG-derived references during rapid workload transitions [34]. In this context, the preservation of negligible mean bias across intensities suggests that multimodal fusion and algorithmic processing effectively maintained central accuracy, even if occasional epoch-level deviations occurred under more demanding conditions. On this basis, secure sensor placement should be acknowledged as a critical determinant of optical signal quality, particularly for ear-worn devices. Prior studies of in-ear PPG have shown that fit and contact pressure strongly influence both accuracy and precision, with poorly seated earbuds exhibiting greater susceptibility to artefact and data loss [37]. In the present study, device fitting was standardised and verified before testing, which likely contributed to the favourable agreement observed. Despite this, small shifts in fit during vigorous exercise remain difficult to eliminate entirely and may account for some of the increased dispersion at higher intensities.

Importantly, these findings align with broader observations across wearable form factors: optical heart-rate monitors tend to perform best when relative motion between sensor and tissue is minimised and when algorithms balance noise suppression against temporal fidelity [16]. The ear canal appears to offer a favourable compromise between signal

stability and user acceptability, but—as with wrist- or arm-based devices—it cannot fully escape the fundamental constraints imposed by movement, tissue deformation, and proprietary signal processing. Accordingly, the observed performance of AirPods Pro 3 reflects the expected behaviour of a well-optimised optical system operating within these physiological and technical boundaries.

Several methodological strengths underpin the interpretability of these findings. First, the study was explicitly designed to evaluate wearable performance under dynamically changing heart-rate conditions, rather than relying on prolonged steady-state averages that can obscure transient error. The graded treadmill protocol incorporated multiple intensity levels and rapid transitions, directly challenging the device in scenarios most relevant to real-world exercise use [16]. Second, agreement was evaluated at the epoch level using a repeated-measures Bland–Altman framework implemented within a mixed-effects model, appropriately accounting for within-participant dependence and temporal structure in the data. Complementary error metrics were calculated using pooled valid Polar–AirPods paired epochs within each intensity category and overall, aligning with contemporary best practice for wearable validation [16]. But these strengths should be interpreted alongside important limitations that define the boundaries of inference. The sample comprised predominantly young, healthy adults, limiting generalisability to older populations, individuals with cardiovascular disease, or those with irregular rhythms. Although a range of Fitzpatrick skin types was represented, darker skin tones (types V–VI) were under-represented, and differential performance across pigmentation cannot be excluded [38,39]. Testing was conducted in a controlled laboratory environment using treadmill-based exercise, which enhances internal validity but may not capture the full spectrum of movements, environmental conditions, and behavioural factors encountered during free-living activity. In addition, heart-rate targets were defined using age-predicted maximal heart rate rather than directly measured maximal effort, introducing imprecision in intensity classification. Finally, as with all consumer wearables, the proprietary nature of signal processing algorithms precluded direct examination of how filtering, averaging, or multimodal fusion influenced responsiveness during rapid heart-rate transitions [36]. Future research should therefore prioritise free-living validation across diverse activity contexts, include older adults and clinical populations, and broader representation of skin pigmentation. Comparative studies examining ear-based sensing alongside wrist- and arm-worn devices within the same protocol, as well as evaluations of responsiveness during high-intensity interval exercise or abrupt workload changes, would further clarify the strengths and limitations of ear-worn heart-rate monitoring. More broadly, the growing integration of consumer wearables into health monitoring raises considerations related to digital access, connectivity, and health and technology literacy. Device performance and utility may be constrained in rural or remote settings, in low-resource environments, or among populations with limited access to smartphones, reliable data infrastructure, or digital health support [40,41]. Future research should therefore prioritise evaluation across more diverse populations and real-world contexts to ensure that the benefits of wearable monitoring are equitably distributed and that findings are not overgeneralised beyond the populations studied.

## Conclusions

In this laboratory-based validation study, AirPods Pro 3 demonstrated close agreement with a chest-worn ECG-derived reference for heart-rate measurement during graded treadmill exercise, with negligible systematic bias and low average error across a wide range of intensities. Variability increased at higher workloads, reflecting expected physiological and biomechanical challenges of optical sensing under dynamic conditions rather than device failure.

## Materials and methods

### Study design and reporting framework

This laboratory-based validation study evaluated agreement between AirPods Pro 3 and a validated criterion heart-rate measure during treadmill exercise. The primary measurement property of interest was agreement under conditions of

dynamically changing heart rate, consistent with the intended use of wearable heart-rate monitoring during physical activity.

The study was designed and reported in accordance with the recommendations of the INTERLIVE consortium for wearable validation studies [16]. Reporting also followed relevant elements of STARD 2015 applicable to continuous measurement comparison [42].

## Ethics

The Human Research Ethics Committee at University College Dublin granted ethical approval on October 17, 2025 (LS-25–74).

## Setting

Data collection took place in a controlled exercise physiology laboratory at University College Dublin, Ireland. The laboratory environment was maintained at a stable ambient temperature of 19–22 °C and was equipped with a motorised treadmill suitable for walking and running protocols. Trained research staff with experience in exercise testing and wearable device setup supervised all sessions. Data were collected between November and December 2025.

## Participants

**Eligibility criteria.** We recruited adults aged 18–65 years who were able to complete treadmill walking and running. Participants were required to be free from diagnosed cardiovascular disease, metabolic conditions that contraindicated exercise testing, and musculoskeletal injuries that could limit exercise participation. We excluded individuals who were pregnant or taking medications known to affect heart rate, including beta-blockers.

**Recruitment and sampling.** Participants were recruited using convenience sampling from the university community and surrounding area. To improve generalisability and reduce spectrum bias, recruitment aimed to include individuals across a range of sex, body mass index, habitual physical activity levels, and skin tone. All participants provided written informed consent before participation.

We derived Fitzpatrick skin type using a questionnaire-based classification algorithm incorporating self-reported skin colour, sunburn response, and tanning ability [43]. Participants were classified into Fitzpatrick skin types I–IV using standard criteria reflecting propensity to burn and ability to tan.

**Sample size considerations.** Sample size was guided by recommendations for agreement studies with repeated measures rather than formal hypothesis testing [16,44]. The target sample size was selected to yield stable estimates of mean bias and 95% limits of agreement with acceptable precision (approximately ±2–3 beats·min⁻¹) across activity conditions.

## Devices and measurement instruments

**Index devices.** The index device was the AirPods Pro 3, a consumer earbud incorporating an in-ear optical heart-rate sensor capable of reporting real-time heart rate during workouts [7]. Heart-rate data were captured via Apple's Fitness and Health frameworks on a paired iPhone. All units were purchased through retail channels. Apple Inc. had no role in the study design, data collection, analysis, or interpretation.

**Criterion reference.** The criterion reference was the Polar H10 chest-worn heart-rate sensor (Polar Electro Oy, Kempele, Finland), a validated chest-worn monitor deriving heart rate from ECG-derived R–R intervals [16,45]. The device derives heart rate from inter-beat (R–R) intervals detected via electrodes secured at the mid-sternal level. Time-stamped R–R interval data exported through the Polar Flow Web app served as the reference against which wearable-derived heart rate was compared.

All AirPods Pro 3 units were operated using identical firmware (version 8B25) and paired to an Apple iPhone running iOS version 26.1. Heart-rate data were recorded during structured workout sessions using Apple's Fitness and Health frameworks as implemented at the time of data collection. The Polar H10 chest strap operated on firmware version 4.0.4 and was paired to a compatible recording device running Polar Flow (version 6.33.1). Data export from the iPhone was performed using Health Auto Export (version 8.3.5) at the highest available sampling frequency. No device firmware or operating system updates occurred during the data collection period.

**Device placement.**  Research staff fitted all devices according to manufacturer instructions. The Polar H10 strap was positioned at the mid-sternal level with moistened electrodes. AirPods Pro 3 were seated securely in the external auditory canal using the manufacturer-recommended ear-tip size to ensure stable contact throughout exercise. Proper fit was confirmed before testing.

### Experimental protocol

**Pre-test standardisation.**  Participants were asked to avoid caffeine and alcohol for 12 hours and vigorous physical activity for 48 hours before testing. All participants completed a PAR-Q+ health screening questionnaire before the laboratory visit.

**Treadmill protocol.**  Each participant completed a single laboratory session lasting approximately 50–60 minutes, including setup, rest, exercise, and recovery. The active treadmill component lasted approximately 40–50 minutes.

The protocol consisted of sequential stages of seated rest, supine rest, walking, running or incline walking, and recovery. Exercise intensity progressed from light to vigorous, targeting 40%, 55%, 65%, 75%, and 85% of age-predicted maximal heart rate ($HR_{max} = 220 - age$). Each steady-state stage lasted three to four minutes and was separated by brief recovery periods. These intensities were selected to span the range of heart rates most relevant to consumer use, while prioritising locomotor activities characterised by repetitive movement patterns, for which wearable heart-rate sensors demonstrate the highest accuracy under laboratory conditions. In accordance with INTERLIVE recommendations, the protocol combined periods sufficient to achieve submaximal steady state with transitions between intensities, thereby challenging device performance under both stable and dynamically changing physiological conditions [16].

Heart-rate values from the AirPods Pro 3 were analysed exactly as provided to end users, without additional filtering, smoothing, or post-processing. This approach was adopted to preserve ecological validity and to evaluate device performance under real-world conditions. Consumer wearable heart-rate estimates reflect proprietary signal processing pipelines that integrate optical signals with motion and contextual data before output; additional investigator-applied filtering would therefore modify the effective measurement and could artificially improve agreement with the reference. In accordance with INTERLIVE recommendations for wearable validation, the index device was evaluated at the output level available to users, thereby ensuring that agreement estimates reflect the accuracy of the device as deployed in practice rather than the performance of post hoc data processing.

For participants uncomfortable with running, a graded walking protocol with increasing incline was used to achieve equivalent heart-rate targets. The protocol also included rapid transitions between activity levels to assess device responsiveness to heart-rate change. Testing was terminated if participants developed exercise-limiting or clinically concerning symptoms (for example, chest pain, dizziness, presyncope, or undue shortness of breath), exceeded 90% of age-predicted HR_max, or requested to stop.

**Data acquisition and synchronisation.**  Device clocks for the paired iPhone and Polar H10 were synchronised to the iPhone system time immediately before each testing session. Recording on both systems was initiated in close temporal proximity (<2 s) and stage transitions were marked contemporaneously using in-application lap or segment markers. Post hoc synchronisation of heart-rate time series was performed using recorded timestamps. Data streams were aligned by nearest-neighbour timestamp matching within each non-overlapping 5-s epoch. For each paired epoch, the absolute time offset between AirPods Pro 3 and Polar H10 measurements was calculated. Epochs in which no corresponding

measurement was available from one device within the same 5-s window were excluded from paired analyses. Stage boundary markers and session logs were used as secondary references to confirm correct temporal correspondence between device recordings and treadmill protocol phases.

## Data processing

**Signal preprocessing.** R–R interval data from the Polar H10 were processed using automated algorithms to identify and remove artefacts and ectopic beats before heart-rate calculation. Brief, isolated deviations of at least 10 beats·min$^{-1}$ between adjacent values that were followed by an immediate return to baseline were classified as transient noise and removed.

Heart-rate values derived from the AirPods Pro 3 were analysed exactly as provided to end users, without additional filtering, smoothing, or post-processing.

**Epoching and exclusions.** To facilitate time-aligned comparison and reduce the influence of short-duration noise, we aggregated both data streams into non-overlapping five-second epochs, in accordance with INTERLIVE recommendations. Within each epoch, heart rate was calculated as the mean of all available measurements for that device.

## Statistical analysis

**Data extraction and preprocessing.** Polar H10 heart-rate data were recorded at one measurement per second and exported via the Polar Flow web application. AirPods Pro 3 heart-rate data were exported at the highest available sampling frequency using the Health Auto Export iOS application (Lybron Sobers). We synchronised device clocks before testing and aligned data streams using timestamps.

Only paired observations occurring within the same five-second epoch were included in the analysis. Polar H10 epochs without a corresponding AirPods Pro epoch were excluded. All analyses were conducted at the epoch level.

**Agreement analysis.** The primary analysis evaluated agreement between AirPods Pro 3 and the criterion measure using a Bland–Altman approach for repeated measures. For each paired epoch, we calculated the difference in heart rate as:

$$difference \ = \ AirPods \ Pro \ 3 \ HR \ - \ Polar \ H10 \ HR$$

Because each participant contributed a time series of observations, measurements were not independent. We therefore modelled differences using a linear mixed-effects model with a fixed intercept representing overall mean bias and a random intercept for participant to account for within-participant dependence. We estimated between-participant and within-participant variance components from the model.

We calculated total variance as the sum of these components and defined the 95% limits of agreement as the mean bias ±1.96 times the square root of the total variance. Where appropriate, we included an autoregressive correlation structure to account for temporal autocorrelation within heart-rate time series. We generated Bland–Altman plots using epoch-level data to visually assess agreement and examine patterns in bias across the heart-rate range.

**Secondary analyses by exercise intensity.** As a secondary analysis, we explored agreement across exercise intensities defined using criterion heart rate expressed as a percentage of age-predicted $HR_{max}$. We classified intensity as rest (<40% HR_max), light (40–59% HR_max), moderate (60–74% HR_max), or vigorous (≥75% HR_max).

Within each intensity category, we examined trends in agreement using absolute error metrics. Mean absolute error (MAE) and mean absolute percentage error (MAPE) were first calculated separately for each participant using epoch-level data and then averaged across participants to avoid disproportionate weighting of individuals with more observations. In addition, participant-level mean signed error was calculated descriptively within each intensity category to characterise directional bias.

 

## Absolute error metrics

To summarise error, we calculated MAE and MAPE from all valid paired Polar–AirPods epochs within each intensity category and across the full dataset.

## Software

All analyses were conducted in Python (version 3.13) using the pandas, NumPy and Matplotlib libraries. The full analysis code is publicly available at github.com/rorylambe/airpods. The minimal anonymized dataset necessary to replicate the findings reported in this study, together with accompanying documentation, is publicly available in the Open Science Framework (OSF) repository: https://doi.org/10.17605/OSF.IO/D4JNS

## Acknowledgments

The funders had no role in study design, data collection and analysis, decision to publish, or preparation of the manuscript.

## Author contributions

**Conceptualization:** Cailbhe Doherty, David Burke, Michael Fuchs, Rory Lambe.

**Data curation:** Cailbhe Doherty, Owen Mitchell, Jasman Brar, Michael Fuchs, Rory Lambe.

**Formal analysis:** Cailbhe Doherty, Rory Lambe.

**Funding acquisition:** Cailbhe Doherty, David Burke.

**Investigation:** Cailbhe Doherty, Owen Mitchell, David Lawless, Jasman Brar, Michael Fuchs.

**Methodology:** Cailbhe Doherty, Owen Mitchell, David Lawless, Jasman Brar, Michael Fuchs.

**Project administration:** Cailbhe Doherty, David Lawless, Jasman Brar, Michael Fuchs, Rory Lambe.

**Resources:** Cailbhe Doherty.

**Software:** Cailbhe Doherty, Rory Lambe.

**Supervision:** Cailbhe Doherty, David Burke, Rory Lambe.

**Validation:** Cailbhe Doherty, Owen Mitchell, David Lawless, Jasman Brar, Michael Fuchs, Rory Lambe.

**Visualization:** Cailbhe Doherty, Rory Lambe.

**Writing – original draft:** Cailbhe Doherty, David Burke, Rory Lambe.

**Writing – review & editing:** Cailbhe Doherty, David Burke, Rory Lambe.

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
