## [Editor Report · Decision Letter 0]

24 Feb 2026

PDIG-D-26-00136Accuracy of heart rate measurement using AirPods Pro (3rd generation) during graded treadmill exercise: a laboratory-based validation studyPLOS Digital Health Dear Dr. Doherty, Thank you for submitting your manuscript to PLOS Digital Health. After careful consideration, we feel that it has merit but does not fully meet PLOS Digital Health's publication criteria as it currently stands. Therefore, we invite you to submit a revised version of the manuscript that addresses the points raised during the review process. Please submit your revised manuscript by Mar 26 2026 11:59PM. If you will need more time than this to complete your revisions, please reply to this message or contact the journal office at digitalhealth@plos.org.  Please include the following items when submitting your revised manuscript:* A letter that responds to each point raised by the editor and reviewer(s). You should upload this letter as a separate file labeled 'Response to Reviewers'. This file does not need to include responses to any formatting updates and technical items listed in the 'Journal Requirements' section below.'. This file does not need to include responses to any formatting updates and technical items listed in the 'Journal Requirements' section below.* A marked-up copy of your manuscript that highlights changes made to the original version. You should upload this as a separate file labeled 'Revised Manuscript with Track Changes'.'.* An unmarked version of your revised paper without tracked changes. You should upload this as a separate file labeled 'Manuscript'.'. If you would like to make changes to your financial disclosure, competing interests statement, or data availability statement, please make these updates within the submission form at the time of resubmission. Guidelines for resubmitting your figure files are available below the reviewer comments at the end of this letter. We look forward to receiving your revised manuscript. Kind regards, Samarra TobyGuest EditorPLOS Digital Health Nicole Li-JessenSection EditorPLOS Digital Health Leo Anthony CeliEditor-in-ChiefPLOS Digital Healthorcid.org/0000-0001-6712-6626  **Journal Requirements:**

i. State the initials, alongside each funding source, of each author to receive each grant. For example: "This work was supported by the National Institutes of Health (####### to AM; ###### to CJ) and the National Science Foundation (###### to AM)."

2. Please ensure that your Ethics Statement is available in its entirety at the beginning of your Methods section, under a subheading 'Ethics Statement'. It must include:

i) The name(s) of the Institutional Review Board(s) or Ethics Committee(s)

ii) The approval number(s), or a statement that approval was granted by the named board(s)

iii) (for human participants or donors) - A statement that formal consent was obtained (must state whether verbal/written) OR the reason consent was not obtained (e.g., anonymity).

3. Please upload separate figure files in .tif or .eps format. Also, remove the figures from your manuscript file but keep the legends.

 If the reviewer comments include a recommendation to cite specific previously published works, please review and evaluate these publications to determine whether they are relevant and should be cited. There is no requirement to cite these works unless the editor has indicated otherwise.  **Additional Editor Comments (if provided):** This is a great project and interesting data and as the use of wearables from consumer to medical grade devices will increase globally over coming years, and more applications are integrated into such devices, these studies are very important. Congratulations to the team on this research.

Minor revisions:

- At 476 Data Processing – further expand on rationale for the AirPods pro heart rate values analysis to further explore rationale if aim is to process as raw data so to speak with no filtration for AirPods.

Methodological Transparency -

- Is there any further information regarding a literature review or specifics around key word or search strategy such as systematically searching PubMed, MEDLINE, Embase, Web of Science, Global Health, Cochrane Library, APA PsycINFO, and APA PsycArticles etc for studies published?

Formatting and Layout Clarification –

- Reformatting the layout of the research article to ensure that Search Strategy relating to any literature review and references, Quality Assessment and Materials and Methods is outlined prior to the Discussion and Conclusion statement to align with publishing formats.

- Additionally ensure referencing formatting is in line with PLOS Digital Health guidelines.

Discussion Considerations (pending word limit considerations):

- Consider further additional commentary regarding limitations of cohort size, demographic and in particular Fitzpatrick skin types to ensure future research is considered regarding cultural and genetic factors that may impact heart rate detection.

- As the potential users continue to grow, wearables will have more sociological and cultural impact in the future. Consideration to expand on the current comments regarding cohort and future studies may be useful to support exploration of technical development, health infrastructure, digital connectivity and digital and technology health equity resources. In relation to future research, the most vulnerable or most affected populations who may have health and technology disparity in access would need to be considered to ensure broad generalisations are not made regarding the application of this technology.

- Furthermore considerations regarding interpretation of clinical bias for future research study designs, in relation to the data regarding age predicted maximal heart rate and careful interpretation to assess what this may be for different cultural, and patient cohorts - this may affect how the data is interpreted and therefore how it is used by consumers or health professionals.

For example in Indigenous populations in other countries, and countries such as Australia with higher rates of cardiovascular and rheumatic heart disease there is a higher risk of cardiovascular disease including arrhythmias, or if there are considerations regarding genetically predisposed cohorts with cardiac conditions that future studies may need to be mindful of regarding what is considered "normal" for that particular cohort.

Additionally, in the discussion section for future research considerations regarding the applicability and results of the use of the technology in rural and remote health care settings, and in particular technology and health literacy to ensure these results are not widely applied without further research and highlighting challenges for populations in rural and remote communities regarding connectivity and digital satellite access may impact technology use and results, in addition to the reliability of the data and its use.**Reviewers' Comments:**    **Figure resubmission:** While revising your submission, we strongly recommend that you use PLOS’s NAAS tool (https://ngplosjournals.pagemajik.ai/artanalysis) to test your figure files. NAAS can convert your figure files to the TIFF file type and meet basic requirements (such as print size, resolution), or provide you with a report on issues that do not meet our requirements and that NAAS cannot fix.

After uploading your figures to PLOS’s NAAS tool - https://ngplosjournals.pagemajik.ai/artanalysis, NAAS will process the files provided and display the results in the "Uploaded Files" section of the page as the processing is complete. If the uploaded figures meet our requirements (or NAAS is able to fix the files to meet our requirements), the figure will be marked as "fixed" above. If NAAS is unable to fix the files, a red "failed" label will appear above. When NAAS has confirmed that the figure files meet our requirements, please download the file via the download option, and include these NAAS processed figure files when submitting your revised manuscript. **Reproducibility:** To enhance the reproducibility of your results, we recommend that authors of applicable studies deposit laboratory protocols in protocols.io, where a protocol can be assigned its own identifier (DOI) such that it can be cited independently in the future. Additionally, PLOS ONE offers an option to publish peer-reviewed clinical study protocols. Read more information on sharing protocols at https://plos.org/protocols?utm_medium=editorial-email&utm_source=authorletters&utm_campaign=protocols

---

## [Editor Report · Decision Letter 1]

1 Apr 2026

Accuracy of heart rate measurement using AirPods Pro (3rd generation) during graded treadmill exercise: a laboratory-based validation study

PDIG-D-26-00136R1

Dear Cailbhe Doherty,

We are pleased to inform you that your manuscript 'Accuracy of heart rate measurement using AirPods Pro (3rd generation) during graded treadmill exercise: a laboratory-based validation study' has been provisionally accepted for publication in PLOS Digital Health.

Best regards,

Cleva Villanueva, M.D., Ph.D.

Academic Editor

PLOS Digital Health

**Additional Editor Comments (if provided):**

The authors have adequately addressed all reviewer comments and revised the manuscript accordingly. Therefore, this editor has decided to accept the manuscript, as it now complies with the requirements of PLOS Digital Health